# Surgical informed consent practice and associated factors among adult postoperative patients in public hospitals of Mekelle, Tigray, Ethiopia 2023/2024. Cross sectional

Fiseha Abadi Gebreanenia[1]*, Hailemarim Berhe Kahsay[2], Desta Siyoum Belay[2], Binyam Gebrehiwet Tesfay[3], Fissha Brhane Mesele[4], Mamush Gidey Abirha[3]

1 Department of Nursing, College of Health Science, Aksum University, Axum, Ethiopia, 2 College of Health Science, School of Nursing, Mekelle University, Mekelle, Ethiopia, 3 Department of Nursing, College of Medicine and Health Science, Adigrat University, Adigrat, Ethiopia, 4 Gambella Teachers' Education and Health Science College, Gambela, Ethiopia

* Fiseha@aku.edu.et

**Data Availability Statement:** All relevant data are within the manuscript and its Supporting Information files.

## Abstract

### Background

Substantial weaknesses and omissions of surgical informed consent are evident and the current elements of the surgical informed consent process are largely neglected in daily practice. This study aimed to assess surgical informed consent practice and associated factors among adult postoperative patients in public hospitals of Mekelle, Tigray, Ethiopia.

### Methods

Institution based cross-sectional study was conducted among 314 adult postoperative patients in public hospitals of Mekelle, Tigray, Ethiopia. Participants were selected using systematic random sampling. A pretested interviewer-administered questionnaire was used to collect data. Descriptive, Descriptive, bivariate and multivariable logistic regression analyses were performed using statistical package for social science version 27.

### Result

Only 35.8% (CI 95%, 30.6, 41) of the respondents were identified to have received the recommended (6 or more) components of surgical informed consent. Educational level [AOR 5.76 (1.02, 32.6)], timing of surgical informed consent delivery [AOR 3.27 (1.5, 7.11)], qualification of counselor who took surgical informed consent them [AOR3.185 (1.21, 8.38)], hospital type [AOR 2.85 (1.26, 6.46)], and duration of counseling [AOR 6.9 (3.33, 14.3)] were statistically significant at P<0.05.

### Conclusion

Majority of participants did not receive comprehensive information during the surgical informed consent process in the study hospitals. To improve the delivery it is suggested that health professionals; create rapport with patient, spend more time during counseling.

**Funding:** The author(s) received no specific funding for this work.

**Competing interests:** This journal is very preferable because of its quality paper publishing especially in health-related researches.

**Abbreviations:** AOR, Adjusted Odds Ratio; CRD, Crude; Odds, Ratio; E.C-, Ethiopian calendar; EMA-, Ethiopian Medical Association; G.C-, Gregorian calendar; GH, General Hospital; IRB, Institutional Review Board; MCC, Motivated Competent Compassionate; MGH, Mekelle General Hospital; OR, Odds Ratio; SIC, Surgical Informed Consent; SPSS, Statically Package Software for Social Science; UK, United Kingdom.

# Introduction

Informed consent is the process in which a health care provider educates a patient about the risks, benefits, and alternatives of a given procedure or intervention. The patient must be competent to make a voluntary decision about whether to undergo the procedure or intervention. Informed consent is both an ethical and legal obligation of medical practitioners and originates from the patient's right to direct what happens to their body [1].

As part of Informed consent, surgical informed consent (SIC) is the process whereby patients are informed of all the necessary information about the general health, type of surgery, subsequent treatment plans, risk and alternatives. Informed consent is built by competent patient to make decision about one's surgical procedure with enough information which is clear, understandable and valid [2, 3].

Surgical informed consent is improperly practiced and violated in many occasions; this is mostly due to lack of health care provider awareness and experience with SIC, heavy workload and negligence of health care providers, and lack of a standard consent form [4, 5]. While evidence suggests that every surgical patient should be informed by the performing healthcare provider about the proposed surgery before signing a consent form, this isn't always the case in clinical settings, particularly in developing countries [6].

A study in Italy revealed that 84.5% of the participants either personally or through a delegate signed the informed consent. Most attention was given to the diagnosis and the type of surgical procedure, which was communicated respectively to 92.8 and 88.2% of the patients [7].

Another study conducted in Nigeria among surgeons revealed that, 54.9% of the surgeons agreed that sufficient information is not provided to patients while obtaining their consent for surgical procedures [8].

In Ethiopia a cross-sectional study conducted on Surgical informed consent using 13 components of SIC showed that, significant proportion (73.5%) of the respondents hasn't received at least six of the 13 components of SIC suggested by the investigators which is unacceptable [12]. Similar study conducted in Addis Ababa among patients undergone caesarian section 44.9% received the minimum required components of surgical informed consent [9, 10]. This could be severe in war torn Tigray as Only 9.7% of health centers, 43.8% of general hospitals and 21.7% of primary hospitals are functional and many health professionals fled out of the region [11].

If adequate information is not provided, the legal consequences could potentially involve lawsuit, professional licensure actions, and negative accreditation implications for the provider and the institution wherein the treatment occurred [12].

According to previous studies factors like educational level, type of surgery, time of counseling, schedule of surgery and duration of counseling were significantly associated with SIC practice [4, 9, 10, 13].

To alleviate the problem the ministry of health of Ethiopia has launched and formulated strategies for healthcare ethics education for students and healthcare professional's development, this program is called 'Motivated, Competent and Compassionate Health Workforce' [14]. However, few health care professionals were trained and its applicability on the ground has been poor [15].

While several studies in Ethiopia have examined patients' knowledge and perception of surgical informed consent, few have investigated the actual practice of informed consent as experienced by patients. Additionally, existing research has primarily focused on specific specialties and among healthcare professionals. In addition to this there is no study conducted in the study area and in the region as well. This study aims to address this gap by investigating

surgical informed consent practice and its associated factors from patient perspective among adult surgical patients in Mekelle public hospitals.

This study's findings will benefit participants by raising their awareness of legal proceedings regarding their treatment. Healthcare providers will gain insight into their current SIC practices and can use this information to improve their services, ultimately minimizing litigation and medico-legal issues. Regulators and ethical committees can utilize this study's findings to assess current SIC practices and identify potential gaps. Additionally, the research can serve as valuable baseline data for future studies with similar interests.

## Methods and materials

### Study design, population, area and period

A cross-sectional institutional study was conducted among 314 postoperative patients in three public hospitals located in Mekelle, the capital city of Tigray, Regional State of Ethiopia. The study hospitals experiences approximately 1001 major surgeries monthly. Data were collected from December 20, 2023, to January 20, 2024. The study population comprised all adult postoperative patients admitted to these hospitals during the specified period.

### Eligibility criteria

**Inclusion criteria.**    All adult patients who underwent surgery and were admitted to public hospitals in Mekelle during the study period were included.

**Exclusion criteria.**    Patients were excluded from the study if they met any of the following criteria:

- Lack of consciousness during data collection.

- Inability to obtain informed consent from a relative.

- Undergoing multiple surgeries during the study period.

**Sampling procedure and technique.**    All public hospitals in Mekelle city offering surgical services were selected for the study. Within each hospital, the sample size for each surgical specialty was proportionally allocated based on the average monthly number of major surgeries performed (N). Participants were chosen using systematic random sampling with a predetermined sampling interval (K) which is three. The K value was determined by dividing the number of average surgeries done in all public hospital to the sample size. In selected hospitals and wards, participants were interviewed based on their sequence of operations done for each ward every K interval (Fig 1).

**Data collection method and tools.**    Data was collected using a structured interviewer-administered questionnaire developed based on established guidelines. A team of nursing students and a supervisor, independent of the study hospitals, gathered data on socio-demographic, service-related characteristics, and essential components of informed consent from postoperative patients over a month, starting on the first postoperative day and continuing until discharge.

### Variables

**Dependent variables.**
➢ Surgical informed consent practice

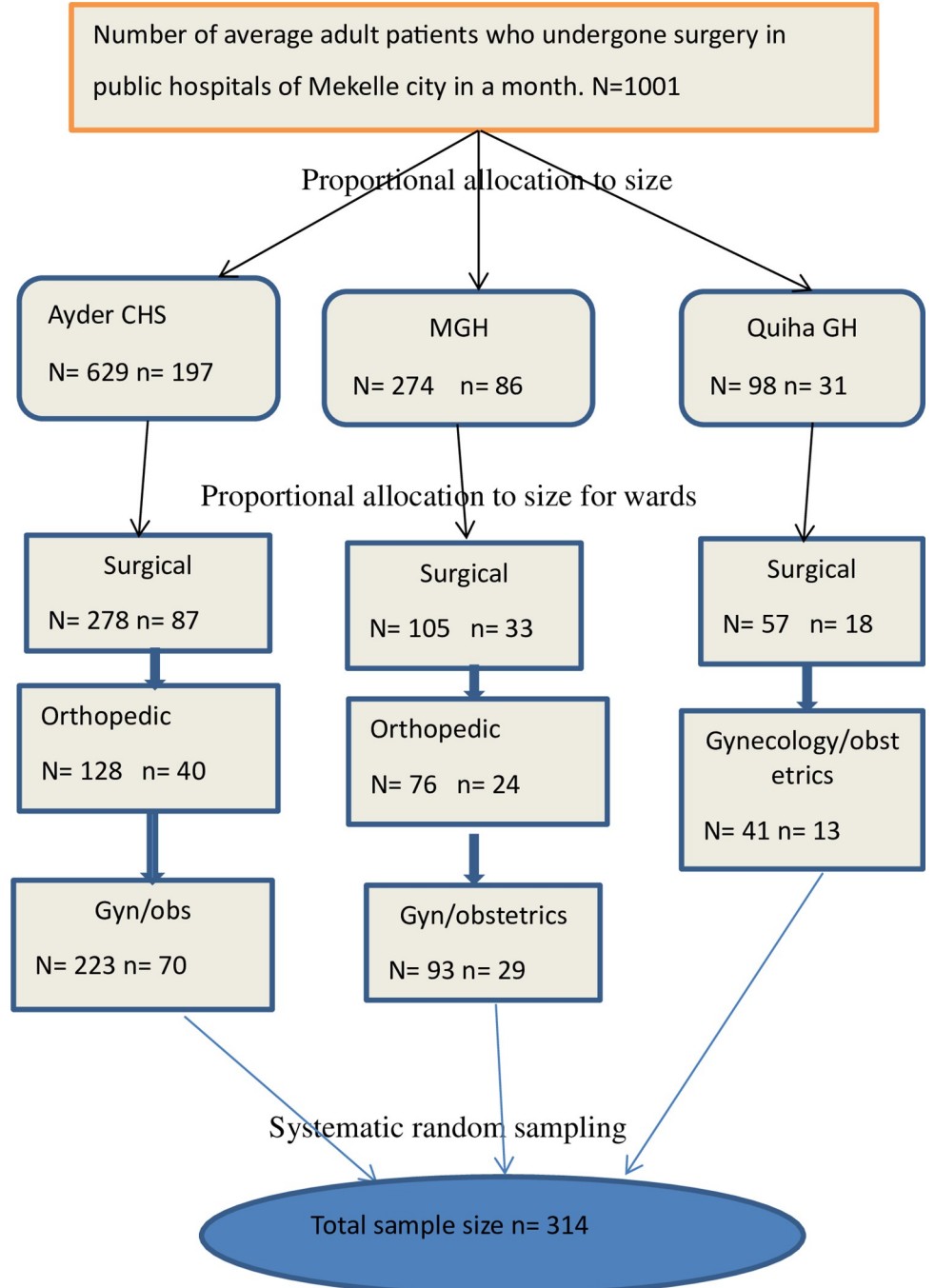

**Fig 1. Schematic presentation of sampling procedure of surgical informed consent practice and associated factors among adult postoperative patients in public hospitals, Meklle, Tigray, Ethiopia 2023/2024.** NB: N total number of surgeries in a month, n proportional size.

**Independent variable.**
➢ Socio demographic characteristic (age, sex, marital status, educational level etc. . .
➢ Service-related characteristics (schedule of surgery, type of hospital, timing of SIC. . .

**Statistical analysis and management.** The data was coded and entered in to Epi Info version 7.2. After cross-checking for consistency and accuracy, the data was cleaned and exported to SPSS version 27 for statistical analysis. Descriptive statistics were performed, Frequency distributions and percentages were employed for categorical variables and the results were summarized using tables, text, and figures.

Both bivariable and multivariable logistic regression analyses were used to determine the association between independent variable and dependent variable. In Bivariable analysis variables having p value less than 0.2 were entered in to multivariable logistic regression to assess statistical association between the outcome variable and independent variables. In multivariable logistic regression, p value <0.05 with 95% confidence interval (CI) were considered statistically significant. The Hosmer- Lemeshow goodness-of-fit model coefficients tests procedure was used to test for model fitting. Kolmogorov-Smirnov test was used to determine normality of data distribution.

**Data quality assurance.** To ensure data quality, a questionnaire was adapted from published literature, translated, and pretested. Data collectors received training and were supervised during data collection. Completed questionnaires were checked for errors. Reliability was assessed using inter-rater reliability and Cronbach's alpha, while face validity was evaluated.

## Ethical considerations

Official ethical clearance letter was obtained from the institutional Review board of the College of Health Sciences of the Mekelle University, after approval the necessary communication were made with chief clinical directors, medical directors and the hospital administrators of each hospital. Written informed consent were obtained from the respective personnel before starting study. Filled questionnaires kept securely and only accessible to the researcher. All methods were performed in accordance with declarations of Helsinki.

## Result

In this study, a total of 310 out of 314 samples were included, yielding a response rate of 98.7%.

### Socio demographic characteristics of study participants

Out of the 310 post-operative patients interviewed, 160 (51.6%) were males. From the total respondents 140 (45.2%) were between the ages of 26 and 40 years. Out of the total study participants (87.4%) were Orthodox followers and (72.3%) of them were married. See details on (Fig 2), (Table 1), (S1 Data set).

### Service-related characteristics of the respondents

Nearly half of the patients (49.4%) who underwent surgery were unaware of qualifications of the healthcare provider who counseled them about surgical informed consent (SIC). The counseling session duration for most patients (77.7%) was less than 5 minutes. Over half of the surgeries were emergency procedures (59.4%). Out of the total surgeries done (43.5%) were general surgery. Additionally, out of the total surgeries in this study 194(62.6%) were performed at Ayder Specialized Hospital Table 2.

### Components of surgical informed consent received by respondents

Almost all (99%) of the respondents or their family members signed an informed consent form. Most respondents (87.7%) were informed about the indication for the surgery. However,

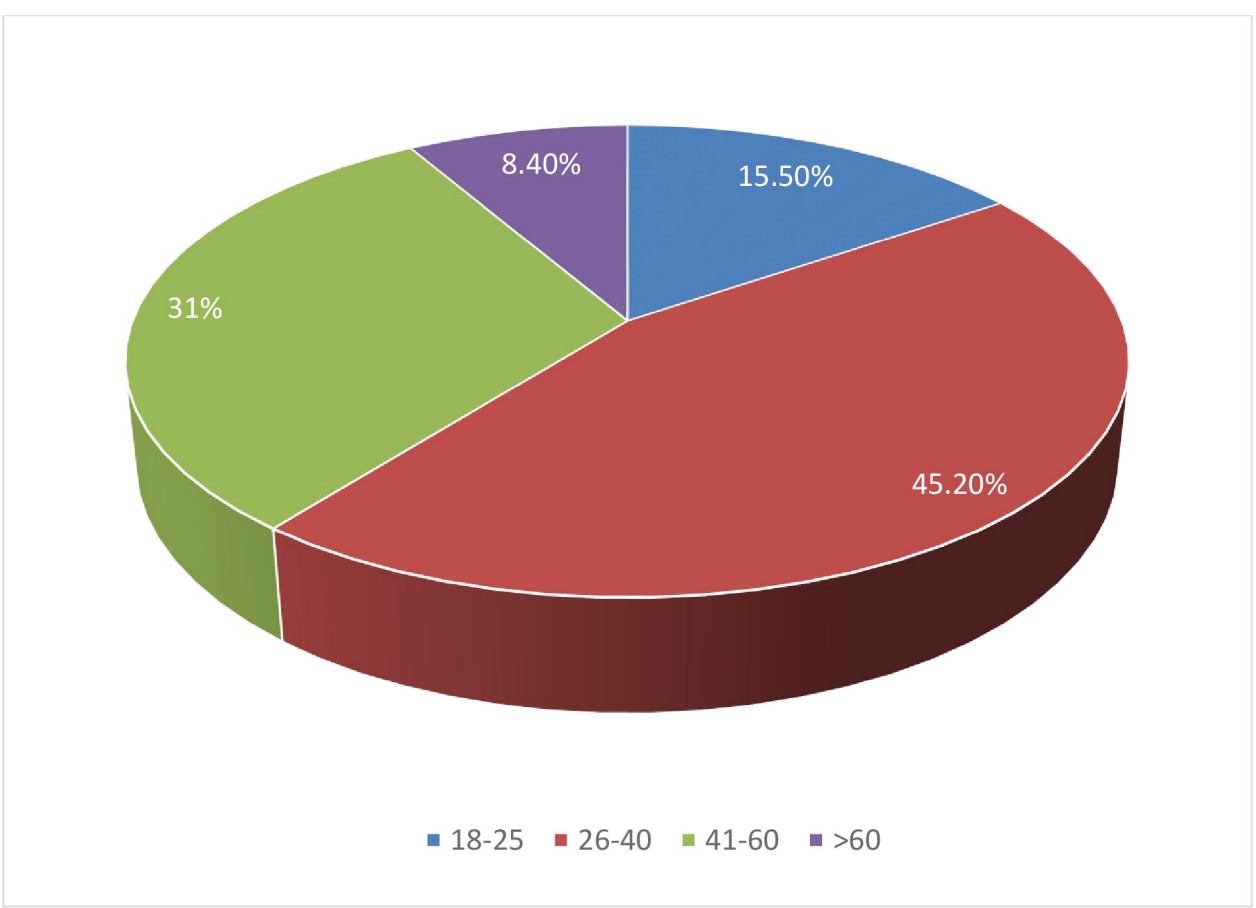

**Fig 2. Age distribution of study participants to assess surgical informed consent practice and associated factors among adult postoperative patients in public hospitals, Meklle, Tigray, Ethiopia 2023/2024.**

only 8.4% of respondents were informed about the expected surgery duration, and very few respondents (1%) were given a chance to choose their anesthesia. More than half (55.5%) of respondents received information about potential complications during the procedure Table 3.

### Receipt of six or more components of surgical informed consent

Based on the total numbers of the components of SIC respondents received before their surgery, out of the total thirteen components 35.8% (CI = 95%, 30.6, 41) of the respondents were identified to have received six or more (the minimum recommendation) components of surgical informed consent (Fig 3).

### Factors associated with receipt of six or more components of surgical informed consent

In Bivariable analysis variables having p value less than 0.2 were entered in to multivariable logistic regression to assess statistical association between the outcome variable and independent variables. In bi-variable regression analysis variables like educational level, occupation, sex, type of anesthesia, type of surgery, schedule of surgery, timing of SIC delivery, health care provider who took SIC, hospital type, and counselor time spent on informed consent were

**Table 1. Socio-demographic characteristics of respondents to assess SIC practice among adult postoperative patients in public hospitals of Mekelle, Ethiopia 2023/2024 (n = 310).**

| Variable | Category | Frequency | % |
|---|---|---|---|
| Sex | Male | 160 | 51.6 |
| | Female | 150 | 48.4 |
| | Total | 310 | 100 |
| Religion | Orthodox | 271 | 87.4 |
| | Muslim | 36 | 11.6 |
| | Catholic | 2 | 0.6 |
| | Protestant | 1 | 0.3 |
| | Total | 310 | 100 |
| Marital status | Married | 224 | 72.3 |
| | Single | 49 | 15.8 |
| | Divorced | 19 | 6.1 |
| | Widowed | 15 | 4.8 |
| | Total | 310 | 100 |
| Educational level | Unable to read and write | 92 | 29.7 |
| | Able to read and write | 47 | 15.2 |
| | Primary (1–8) | 54 | 17.4 |
| | Secondary (9–12) | 64 | 20.6 |
| | Diploma | 27 | 8.7 |
| | Degree | 23 | 7.4 |
| | Master and above | 3 | 1 |
| | Total | 310 | 100 |
| Occupation | Government employee | 45 | 14.5 |
| | Private employee | 18 | 5.8 |
| | Farmer | 114 | 36.8 |
| | Housewife | 72 | 23.2 |
| | Private business | 23 | 7.4 |
| | Daily laborer | 13 | 4.2 |
| | Other | 25 | 8.1 |
| | Total | 310 | 100 |
| Residence | Urban | 146 | 47.1 |
| | Rural | 164 | 52.9 |
| | Total | 310 | 100 |

significant at P = <0.2, thus they were selected for multi-variable regression analysis. Following multi-variable regression analysis variables like educational level, timing of SIC delivery, Qualification of counselor who took SIC, hospital type, and counselor time spent on informed consent were significant at P<0.05 Table 4.

## Discussion

The objective of this study was to evaluate the practice of SIC based on the recommended 13 components. The study mainly focuses on patients admitted to surgical, gynecology/obstetrics and orthopedics wards as there are increased claims of malpractice in these areas [16].

The receipt of six or more (the minimum recommendation) components of SIC in this study is (35.8%) with (CI 95%, 30.6, 41), which is slightly higher than the study conducted at Hawassa (26.5%) [9]. This difference might be attributable to the study area covered, which exposes patients to various health professionals and different working environments. This is

**Table 2. Service-related characteristics of the respondents to assess SIC practice among adult postoperative patients in public hospitals of Mekelle, Ethiopia 2023/2024 (n = 310).**

| Variable | Category | Frequency | Percent |
|---|---|---|---|
| Duration of counseling | <05 minutes | 241 | 77.7 |
| | >05 minutes | 69 | 22.3 |
| | Total | 310 | 100 |
| Qualification of counselor who took surgical informed consent | Senior physician | 68 | 2 |
| | Resident physician | 29 | 9.4 |
| | Intern or GP | 5 | 1.6 |
| | Nurse | 32 | 10.3 |
| | Midwife | 23 | 7.4 |
| | Did not know | 153 | 49.4 |
| | Total | 310 | 100 |
| Schedule of surgery | Elective | 126 | 40.6 |
| | Emergency | 184 | 59.4 |
| | Total | 310 | 100 |
| Type of hospital | Specialized | 194 | 62.6 |
| | General | 116 | 37.4 |
| | Total | 310 | 100 |
| Type of anesthesia | General | 137 | 44.2 |
| | Spinal | 173 | 55.8 |
| | Total | 310 | 100 |
| Timing of counseling for informed consent | The day before date of surgery | 109 | 35.2 |
| | On the day of surgery | 58 | 18.7 |
| | Immediately before surgery | 126 | 40.6 |
| | On the operation table | 17 | 5.5 |
| | Total | 310 | 100 |
| Type of surgery | General surgery | 135 | 43.5 |
| | Orthopedic | 64 | 20.6 |
| | Gynecology/obstetric | 111 | 35.9 |
| | Total | 310 | 100 |

because this study included all public hospitals and post-op wards, whereas the latter study was exclusively conducted in the gynecology/obstetrics department of a specific hospital. On the other hand, it is lower than the study conducted in Addis Ababa (44.9%) [10]. This disparity is likely due to decreased staff morale in helping patients in the study area, possibly stemming from a lack of incentives, service delivery arrangements, resource capacities and increased workload [5].

In this study, 88% of participants were informed about the indication for the proposed surgery. While this is commendable, it should ideally reach 100% as it serves as a foundation for understanding other details. Interestingly, the findings are similar to studies conducted in Hawassa, Italy, and Addis Ababa teaching hospitals, with respective rates of 87%, 88%, and 89% [7, 9, 10].

Findings of this study shows majority of the patient's didn't get enough information on, options of alternative treatments, the type of health professional who deliver SIC, type of anesthesia to be used, complications of surgery, given an opportunity to ask, benefits and duration of the surgery. This is comparable to the study conducted in St. Paul's Hospital Millennium Medical College, China and Nepal military hospital [17–19]. This shows poor practice of SIC

**Table 3. Essential components of surgical informed consent received by respondents to assess SIC practice among adult postoperative patients in public hospitals of Mekelle, Ethiopia 2023/2024 (n = 310).**

| Variables | Responses, n, % | | |
|---|---|---|---|
| | **Yes** | **No** | **Do not remember** |
| Respondent/respondent's family was requested for surgical informed consent | 303 (97.7) | 5(1.6) | 2(0.6) |
| Respondent/respondent's family signed on a surgical informed consent form | 307(99) | 3(1) | _ |
| Respondent was informed why the surgery will be performed (indication of surgery) | 272 (87.8) | 33(10.6) | 5(1.6) |
| Respondent was informed the expected time the surgery will take | 26(8.4) | 273 (88.1) | 11(3.5) |
| Respondent was informed about presence/absence of alternative treatment options | 47(15.2) | 254 (81.9) | 9(2.9) |
| Respondent was informed about type of anesthesia to be used | 19(6.1) | 275 (88.7) | 16(5.2) |
| Respondent was given counseling aids which assist in decision making | 140 (45.2) | 163 (52.6) | 7(2.2) |
| Respondent was informed about potential complication/s which may arise | 203 (65.5) | 100 (32.3) | 7(2.2) |
| Respondent was informed about consequences of refusing the proposed surgery | 179 (57.7) | 125 (40.3) | 6(2) |
| There was a favorable environment to say "No" to the proposed surgery | 58(18.7) | 248(80) | 4(1.3) |
| Respondent was given adequate time for decision to sign on the informed consent form | 164 (52.9) | 139 (44.8) | 7(2.3) |
| Respondent was given opportunity to choose from anesthesia options | 3(1) | 298 (96.1) | 9(2.9) |
| Respondent was given an opportunity to ask question | 127(41) | 179 (57.7) | 4(1.3) |

by health care providers that may be attributed to less training on SIC components, work load and poor knowledge on SIC and its consequences [5].

In addition, the majority of participants in this study (88.7%, 96%) didn't receive information about the anesthesia to be used and did not get an opportunity to choose from the available anesthesia options. This indicates a poor attitude for shared decision making and right of a patient to participate in their clinical care, informed consent practice is affected by health professionals attitude towards it as evidenced by a study conducted in south eastern Ethiopia [5].

A multi-variable regression analysis identified several factors associated with patients receiving the minimum recommended components of SIC. Participants counseled for more than 5 minutes were almost seven times more likely to receive 6 or more components of SIC than those counseled for less than 5 minutes. This is comparable to the study conducted in Addis Ababa which is two times. These findings suggest that when health professionals devote more time to patients, the delivery of key SIC components significantly increases as evidenced by a study conducted in south eastern Ethiopia [5].

Participants counseled one day before surgery were three times more likely to receive the minimum recommended (6 or more) components of SIC compared to those counseled immediately before surgery. This finding is in line with a study conducted in Hawassa as 80% of patients who received SIC immediately before surgery were less likely to receive 6 or more components of SIC. This shows patients counseled earlier before surgery have adequate time to discuss their surgery, thus they are not rushed to undergo surgery with limited information [9].

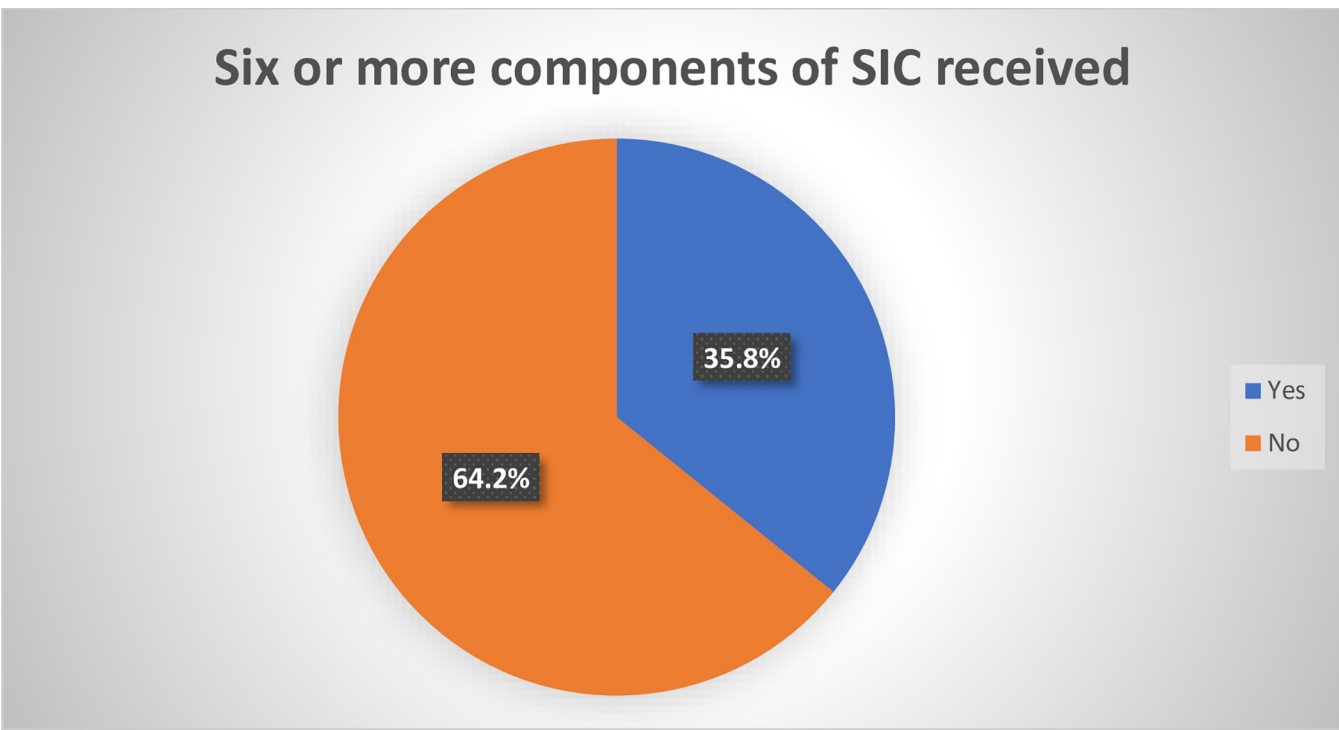

**Fig 3. Receipt of 6 or more components of surgical informed consent, among adult postoperative patients in public hospitals of Mekelle, Tigray, Ethiopia 2023/2024.**

Participants with a degree or higher educational level were 5.7 times more likely to receive 6 or more components than those who were unable to read or. These findings are in line with a study conducted in Iran, and Addis Ababa. This might be due to educated participants tend to ask more questions, which leads them to receive more information about the proposed surgery [10, 13].

Participants who underwent surgery at a specialized hospital were 2.8 times more likely to receive 6 or more components of SIC than those who went to general hospitals, Participants who were counseled by a senior physician are 3 times more likely to receive 6 or more components of SIC than those who didn't know who counseled them. This is may be due to specialized hospital Often have greater staffing and resources, including students who can provide additional information about the surgery, and senior physicians Tend to offer more comprehensive explanations about the procedure and hold primary responsibility for their patients' overall health.

## Strength and limitations of the study

### Strength of the study.

- The study tried to identify factors affecting receipt of the recommended components of SIC in all public hospitals, as existing researches excluded general hospitals.

- This study expands on previous Ethiopian research, which was limited to Obstetrics/Gynecology departments, by encompassing all post-operative wards.

### Limitations of the study.

- Recall bias was a concern as patients were interviewed after surgery. As it was cross-sectional study, doesn't show the cause-and-effect relationship.

**Table 4. Bi-variables and multi-variable analysis to assess factors associated with receipt of six or more components of SIC among adult postoperative patients in public hospitals of Mekelle, Ethiopia 2023/2024 n = 310.**

| Variable | Category | Received 6 or more components of SIC | | COR (95%CI) | AOR (95% CI) | P-value |
|---|---|---|---|---|---|---|
| | | Yes | No | | | |
| Educational level | able to read and write | 16 | 30 | 1.47(0.69,3.15) | 2.29(0.84,6.2) | 0.105 |
| | Primary (1–8) | 19 | 38 | 1.38(0.67,2.82) | 2.1(0.75,5.9) | 0.157 |
| | Secondary (9–12) | 19 | 41 | 1.28(0.63,2.64) | 2.5(0.68,8.2) | 0.129 |
| | Diploma | 15 | 13 | 3.18(1.33,7.62) | 5.1(0.93,28.15) | 0.06 |
| | Degree and above | 17 | 8 | 5.86(2.25,15.27) | **5.76(1.02,32.6)\*** | **0.047\*** |
| | Unable to read and write | 25 | 69 | Ref. | Ref. | |
| Duration of counseling | <05 minutes | 61 | 180 | Ref. | Ref. | |
| | >05 minutes | 50 | 19 | 7.765(4.25,14.19) | **65.9(3.33,14.3)\*** | **<0.001\*** |
| Qualification of counselor who took SIC | Did not know | 48 | 105 | Ref. | Ref. | |
| | Senior physician | 39 | 29 | 2.94(1.63,5.3) | **3.18(1.21,8.38)\*** | **0.019\*** |
| | Resident physician, Intern or GP | 10 | 24 | 0.91(0.4,2) | 0.445(0.15,1.35) | 0.153 |
| | Nurse | 9 | 23 | 0.856(0.37,1.99) | 1.89(0.59,6.1) | 0.287 |
| | Midwife | 5 | 18 | 0.6(0.21,1.73) | 1.34(0.3,6) | 0.704 |
| Type of hospital | General | 31 | 85 | Ref. | Ref. | |
| | Specialized | 80 | 114 | 1.92(1.16,3.17) | **2.85(1.26,6.46)\*** | **0.012\*** |
| Timing of counseling for informed consent | Immediately before surgery | 35 | 108 | Ref. | Ref. | |
| | The day before date of surgery | 59 | 50 | 3.64(2.13,6.22) | **3.27(1.5,7.11)\*** | **0.003\*** |
| | On the day of surgery | 17 | 41 | 1.28(0.65,2.53) | 0.86(0.36,2.1) | 0.751 |
| Occupation | Daily laborer | 4 | 9 | Ref. | Ref. | |
| | Private employee | 6 | 12 | 1.125(0.243,5.2) | 0.51(0.79,3.3) | 0.481 |
| | Farmer | 37 | 77 | 1.08(0.31,3.74) | 1.09(0.21,5.53) | 0.919 |
| | Housewife | 16 | 56 | 0.643(0.175,2.36) | 0.99(0.17,5.6) | 0.988 |
| | Private business | 5 | 18 | 0.625(0.134,2.9) | 0.57(0.85,3.84) | 0.564 |
| | Government employee | 29 | 16 | 4.08(1.08,15.37) | 1.59(0.24,10.5) | 0.629 |
| | Other | 14 | 11 | 2.86(0.69,11.82) | 2.34(0.43,12.8) | 0.326 |
| Sex | Female | 48 | 102 | Ref. | Ref. | |
| | Male | 63 | 97 | 1.38(0.8,2.2) | 0.92(0.37,2.3) | 0.86 |
| Schedule of surgery | Elective | 51 | 75 | 1.4(0.88,2.25) | 1.01(0.49,2.07) | 0.977 |
| | Emergency | 60 | 124 | Ref. | Ref. | |
| Type of surgery | General surgery | 53 | 82 | Ref. | Ref. | |
| | Orthopedic | 30 | 34 | 1.37(0.75,2.5) | 1.28(0.55,2.98) | 0.565 |
| | Gynecology/obstetric | 28 | 83 | 0.52(0.3,0.9) | 0.76(0.24,2.4) | 0.641 |
| Type of anesthesia | General | 59 | 78 | 1.76(1.1,2.81) | 1.07(0.51,2.24) | 0.854 |
| | Spinal | 52 | 121 | Ref. | Ref. | |

\* = Statistically significant with p- value < 0.2   1 = Reference group of the variable

\*\* = Statistically significant with p- value <0.05

## Conclusion

This study showed that majority of participants who underwent both elective and emergency surgery did not receive comprehensive information during the SIC process in the study hospitals. The absence of thorough and standardized pre-operative counseling for surgical procedures undermines, the quality-of-care patients receive and ability of healthcare facilities to

meet patient expectations and information needs. Educational level, timing of SIC delivery, qualification of counselor who took SIC, hospital type, and counselor time spent on informed consent had significant effect on receipt of six or more components of SIC.

To improve SIC practice, ethical committees and regulators should develop standardized SIC forms and monitor their implementation. Hospitals should provide training on informed consent and ensure adherence. Healthcare professionals should deliver comprehensive information, build rapport with patients, and encourage patient participation. Future research should employ observational studies to actively follow the SIC process.

## Supporting information

**S1 Data set.**
(XLSX)

## Acknowledgments

First of all, I would like to express my sincere gratitude to the study participants for their willingness to participate in the study, and to my data collectors and supervisors for their facilitation, organization and collection of the data throughout the study period.

Finally, I extend my sincere thanks to Aksum University for providing me with the opportunity to expand my knowledge and refine my skills. I am also grateful to Mekelle University, College of Health Sciences, and School of Nursing for their support in completing my thesis report.

## Author Contributions

**Conceptualization:** Fiseha Abadi Gebreanenia, Hailemarim Berhe Kahsay, Desta Siyoum Belay.

**Data curation:** Fiseha Abadi Gebreanenia, Binyam Gebrehiwet Tesfay, Fissha Brhane Mesele, Mamush Gidey Abirha.

**Formal analysis:** Fiseha Abadi Gebreanenia, Hailemarim Berhe Kahsay, Desta Siyoum Belay, Fissha Brhane Mesele, Mamush Gidey Abirha.

**Investigation:** Fiseha Abadi Gebreanenia.

**Methodology:** Fiseha Abadi Gebreanenia.

**Project administration:** Fiseha Abadi Gebreanenia.

**Software:** Fiseha Abadi Gebreanenia.

**Supervision:** Fiseha Abadi Gebreanenia.

**Writing – original draft:** Fiseha Abadi Gebreanenia, Binyam Gebrehiwet Tesfay.

**Writing – review & editing:** Fiseha Abadi Gebreanenia, Hailemarim Berhe Kahsay, Desta Siyoum Belay.

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
