## [Editor Report · Decision Letter 0]

25 Jun 2024

PONE-D-24-22632Surgical informed consent practice and associated factors among adult postoperative patients in public hospitals of Mekelle, North Ethiopia 2023/2024. Cross sectionalPLOS ONE

Dear Dr. Gebreanenia,

Thank you for submitting your manuscript to PLOS ONE. After careful consideration, we feel that it has merit but does not fully meet PLOS ONE’s publication criteria as it currently stands. Therefore, we invite you to submit a revised version of the manuscript that addresses the points raised during the review process.

It would be more interesting if Some parts of the methods and materials are in paragraph form. For example, study area, period, design and population can be summarized in one paragraph.

We look forward to receiving your revised manuscript.

Kind regards,

Kahsu Gebrekidan, Ph.D.

Academic Editor

PLOS ONE

Journal Requirements:

"This journal is very preferable because of its quality paper publishing especially in health-related researches."

6. Please amend either the title on the online submission form (via Edit Submission) or the title in the manuscript so that they are identical.

7. Your ethics statement should only appear in the Methods section of your manuscript. If your ethics statement is written in any section besides the Methods, please delete it from any other section. 

8. Please ensure that you refer to Figures 1 and 2 in your text as, if accepted, production will need this reference to link the reader to the figure.

9. Please include a separate caption for each figure in your manuscript.

**Additional Editor Comments:**

Before we proceed with the review process, the manuscript needs revision

1. You need to revise the manuscript based on the authors guideline of PLOS One

2. Some parts of the methods and materials section are not important e.g. Dissemination of the results

3. I would suggest you refer any article published on PLoS one

---

## [Author Response · Author response to Decision Letter 0]

24 Jul 2024

To; Kahsu Gebrekidan, Ph.D.

Academic Editor

PLOS ONE

---

## [Decision Letter · Decision Letter 1]

2 Aug 2024

PONE-D-24-22632R1Surgical informed consent practice and associated factors among adult postoperative patients in public hospitals of Mekelle, North Ethiopia 2023/2024. Cross sectionalPLOS ONE

Dear Dr. Gebreanenia,

Thank you for submitting your manuscript to PLOS ONE. After careful consideration, we feel that it has merit but does not fully meet PLOS ONE’s publication criteria as it currently stands. Therefore, we invite you to submit a revised version of the manuscript that addresses the points raised during the review process.

NB: Comments from the second reviewer attached as a document named ''Reviewer comments''

We look forward to receiving your revised manuscript.

Kind regards,

Kahsu Gebrekidan, Ph.D.

Academic Editor

PLOS ONE

Reviewers' comments:

Reviewer's Responses to Questions

**Comments to the Author**

1. If the authors have adequately addressed your comments raised in a previous round of review and you feel that this manuscript is now acceptable for publication, you may indicate that here to bypass the “Comments to the Author” section, enter your conflict of interest statement in the “Confidential to Editor” section, and submit your "Accept" recommendation.

Reviewer #1: (No Response)

Reviewer #2: (No Response)

2. Is the manuscript technically sound, and do the data support the conclusions?

Reviewer #1: Yes

Reviewer #2: Partly

3. Has the statistical analysis been performed appropriately and rigorously? 

Reviewer #1: Yes

Reviewer #2: N/A

4. Have the authors made all data underlying the findings in their manuscript fully available?

Reviewer #1: Yes

Reviewer #2: Yes

5. Is the manuscript presented in an intelligible fashion and written in standard English?

Reviewer #1: Yes

Reviewer #2: No

6. Review Comments to the Author

Reviewer #1: (No Response)

Reviewer #2: Here are my recommendations about the mentioned MS:

Abstract:

• Writing Keywords.

Introduction:

• Line 49 remove one of the parentheses.

Methodology:

• Change some words study area to the setting of the study, furthermore sampling procedure and technique to sampling etc.

• In setting of the study no need to unnecessary information about the city.

• You can write Study design, setting, period, and sample size all in one paragraph. Furthermore population, population, sample size and sampling etc.

• Present inclusion and exclusion criteria structurally and academically.

• Study tool is not interesting collect and summarize all in one paragraph.

• Study variables also is not interesting collect and summarize all in one paragraph

• No need to operational definitions if needed you can allocate them in Appendix.

• Validity and reliability of tools should exist.

• In data quality assurance how did you determined 5% of the sample size? put the reference.

• Statistical analysis not data analysis

• Dedicate a section for describing tools more clearly.

• Mention the validation and reliability process for tools that have been used in the present study.

• How did you determine the normality of your data?

Results:

• How did you determined to retain variables in to multivariate regression?

Discussion:

• The discussion section needs to be revision and it is also better to discuss the results with available studies and explain the results in a better way.

• There is no need to rewrite results in the discussion section.

Conclusion:

• Looks fine.

References:

• Use new and more references in the introduction and discussion section.

Figures and tables:

• Looks fine.

Some more issues should be considered necessary for publication:

• Suggestions for future studies also be mentioned.

• Please provide at least two related strengths for MS.

• The manuscript need proofreading by a native speaker.

7. PLOS authors have the option to publish the peer review history of their article (what does this mean?). If published, this will include your full peer review and any attached files.

Reviewer #1: No

Reviewer #2: **Yes: **Salar Omar Abdulqadir

ORCID ID (https://orcid.org/0000-0002-4831-9577)

---

## [Author Response · Author response to Decision Letter 1]

30 Aug 2024

Point by point response to reviewer 2

1. I have included the key words.

2. I have removed the repeated parenthesis.

3. I have revised the write up of the study area, population, sample size and procedure.

4. I have written the exclusion and inclusion criteria, structurally and academically. 

5. I have summarized study variable into one paragraph.

6. I have summarized study tool into one paragraph.

7. I have removed the operational definition. 

8. I have written the reliability, normality and validity tools.

9. I used 5% of my sample for pretest by referring to similar studies.

10. I have amended word on the methodology part.

11. I have determined my data normality using SPSS software (by conducting Q_Q plot and Kolmogorov-Smirnov test) .

12. I have determined to retain variables from bivariate to multivariate regression using P-value which is <0.2. 

13. I have revised the discussion part. 

14. I have used recent and available references.

15. I have checked the grammar and flow of my write up using AI.

---

## [Decision Letter · Decision Letter 2]

10 Sep 2024

Surgical informed consent practice and associated factors among adult postoperative patients in public hospitals of Mekelle, North Ethiopia 2023/2024. Cross sectional

PONE-D-24-22632R2

Dear Mr. Fiseha,

We’re pleased to inform you that your manuscript has been judged scientifically suitable for publication and will be formally accepted for publication once it meets all outstanding technical requirements.

Kind regards,

Kahsu Gebrekidan, Ph.D.

Academic Editor

PLOS ONE

Additional Editor Comments (optional):

Reviewers' comments:

Reviewer's Responses to Questions

**Comments to the Author**

1. If the authors have adequately addressed your comments raised in a previous round of review and you feel that this manuscript is now acceptable for publication, you may indicate that here to bypass the “Comments to the Author” section, enter your conflict of interest statement in the “Confidential to Editor” section, and submit your "Accept" recommendation.

Reviewer #1: (No Response)

Reviewer #2: (No Response)

2. Is the manuscript technically sound, and do the data support the conclusions?

Reviewer #1: Yes

Reviewer #2: Yes

3. Has the statistical analysis been performed appropriately and rigorously? 

Reviewer #1: Yes

Reviewer #2: Yes

4. Have the authors made all data underlying the findings in their manuscript fully available?

Reviewer #1: Yes

Reviewer #2: Yes

5. Is the manuscript presented in an intelligible fashion and written in standard English?

Reviewer #1: Yes

Reviewer #2: Yes

6. Review Comments to the Author

Reviewer #1: (No Response)

Reviewer #2: Dear author, thank you for your understanding; however, some issues still need to be addressed.

Here are my recommendations about the mentioned MS:

Abstract:

• Looks good.

Introduction:

• Looks good.

Methodology:

• Cronbach’ alpha value should be existed. Furthermore, tell for which part of the questionnaire you applied.

• How did face validity is conducted? Mention it in the validity and reliability section.

• Please write down the code of ethics you received from the institution you mentioned.

• Describing tools more clearly.

Results:

• Looks good.

Discussion:

• Looks good

Conclusion:

• Looks fine.

References:

• Looks good.

Figures and tables:

• Looks fine.

7. PLOS authors have the option to publish the peer review history of their article (what does this mean?). If published, this will include your full peer review and any attached files.

Reviewer #1: **Yes: **Dr.Nahla Abdulrahman

Reviewer #2: No

---

## [Editor Report · Acceptance letter]

23 Sep 2024

PONE-D-24-22632R2 

PLOS ONE

Dear Dr. Gebreanenia, 

I'm pleased to inform you that your manuscript has been deemed suitable for publication in PLOS ONE. Congratulations! Your manuscript is now being handed over to our production team.

Kind regards, 

on behalf of

Dr. Kahsu Gebrekidan 

Academic Editor

PLOS ONE